# Fractal Dimension Analysis of Melanocytic Nevi and Melanomas in Normal and Polarized Light—A Preliminary Report

**DOI:** 10.3390/life12071008

**Published:** 2022-07-07

**Authors:** Paweł Popecki, Marcin Kozakiewicz, Marcin Ziętek, Kamil Jurczyszyn

**Affiliations:** 1Department of Oral Surgery, Wroclaw Medical University, Krakowska 26, 50-425 Wroclaw, Poland; pawel.popecki@umw.edu.pl (P.P.); kamil.jurczyszyn@umw.edu.pl (K.J.); 2Department of Maxillofacial Surgery, Medical University of Lodz, 113 S. Zeromski Street, 90-549 Lodz, Poland; 3Department of Oncology, Wroclaw Medical University, Plac Hirszfelda 12, 53-413 Wroclaw, Poland; marcin.zietek@umw.edu.pl; 4Department of Surgical Oncology, Wroclaw Comprehensive Cancer Center, Plac Hirszfelda 12, 53-413 Wroclaw, Poland

**Keywords:** fractal dimension analysis, skin pigmented lesions, melanoma, dysplastic nevus, benign nevus

## Abstract

Clinical diagnosis of pigmented lesions can be a challenge in everyday practice. Benign and dysplastic nevi and melanomas may have similar clinical presentations, but completely different prognoses. Fractal dimensions of shape and texture can describe the complexity of the pigmented lesion structure. This study aims to apply fractal dimension analysis to differentiate melanomas, dysplastic nevi, and benign nevi in polarized and non-polarized light. A total of 87 Eighty-four patients with 97 lesions were included in this study. All examined lesions were photographed under polarized and non-polarized light, surgically removed, and examined by a histopathologist to establish the correct diagnosis. The obtained images were then processed and analyzed. Area, perimeter, and fractal dimensions of shape and texture were calculated for all the lesions under polarized and non-polarized light. The fractal dimension of shape in polarized light enables differentiating melanomas, dysplastic nevi, and benign nevi. It also makes it possible to distinguish melanomas from benign and dysplastic nevi under non-polarized light. The fractal dimension of texture allows distinguishing melanomas from benign and dysplastic nevi under polarized light. All examined parameters of shape and texture can be used for developing an automatic computer-aided diagnosis system. Polarized light is superior to non-polarized light for imaging texture details.

## 1. Introduction

The differential diagnosis of pigmented nevi may present itself as a considerable challenge in everyday clinical practice. The majority of pigmented lesions are acquired melanocytic nevi. These benign neoplasms consist of a particular kind of melanocytic cells: larger, devoid of the usually present dendritic processes, organized in nests, called nevus cells [1]. Depending on the location of melanocytes in relation to the dermoepidermal junction, junctional, intradermal, and compound nevi can be distinguished. Clinically, they present as small, dark-colored macules or papules with a uniform pigment distribution and a regular, symmetrical shape [2]. Malignant transformation of a benign nevus, although possible, is very unlikely, so these lesions do not require surgical excision. On the other hand, the presence of a large number of benign nevi is a strong prognostic factor for the development of melanoma [3,4]. A dysplastic nevus is often defined as an intermediate lesion between a benign nevus and a melanoma, occurring in about 2–8% of the population [5]. It was first described in 1978 independently by Clark and Lynch [6,7]. Clinically, it shows an irregular border, non-uniform distribution of the pigment, and a diameter greater than 5 mm [8]. Histological features of dysplastic nevi are atypia of cells, melanocytic hyperplasia, architectural disorders, and concentric or lamellar fibroplasia [9]. Despite the similarity to melanoma in clinical presentation and its histopathological features, progression of dysplastic nevus to melanoma is relatively rare. Still, the presence of even one dysplastic nevus in a patient significantly increases the risk of developing melanoma [10].

Melanoma is a rare type of skin cancer that results from the malignant transformation of melanocytes. Its aggressive clinical course and tendency to metastasize leads to high mortality—it is responsible for 90% of deaths caused by skin cancer but accounts for less than 5% of skin cancer cases [11,12]. The prognosis for this neoplasm depends on the depth of infiltration, tumor ulceration, and the presence of metastases in local lymph nodes and other organs. Despite the high mortality rate of this cancer, detection of melanoma in low stages leads to a 5-year survival rate of more than 90% [13]. The gold standard in the differential diagnosis of melanoma is histopathological examination [14]. Because it requires surgical removal of the lesion, it cannot be used for all pigmented nevi. Each surgical procedure carries the risk of certain complications, leading to scar formation, which may cause aesthetic or even functional problems. Additionally, even if performed on all lesions, it would not reduce the risk of developing new nevi and de novo melanoma. Therefore, it is necessary to use non-invasive methods to diagnose pigmented lesions.

Clinical diagnosis of pigmented lesions is based mainly on visual examination of the lesion, most often using a dermatoscope. This hand-held instrument provides magnification ranging from ×10 to ×100 and a light source, which allows for imaging deeper layers of the skin, invisible in examination with the naked eye [14]. Examination in polarized light expands the capabilities of this method. Polarized dermoscopy reduces the possibility of reflection, which leads to visualizing even deeper layers of lesion than in non-polarized light [15]. This leads to different contrast and therefore to different image segmentation results. Features of the pigmented lesion assessed during dermoscopy include color distribution, vascular patterns, pigment network, presence of structures like dots, globules, streaks, blotches, veils, fissures, ridges, pseudopods, and regression patterns. Benign lesions usually show regular pigment network without interruptions, brown globules, homogeneous colors, symmetric appearance, regular, well-defined borders. Dysplastic nevi usually present irregular pigment network with interruptions, heterogeneous colors, and irregular border [16]. Features characteristic of melanoma are heterogenicity in colors and structures, asymmetry of pigment distribution and shape, irregular border with fading margins, structureless areas, gray-blue or whitish veils, pseudopods, radial streaming, and atypical vascular patterns [16,17]. Due to the better visualization of features and irregularities of the examined structures, the use of dermoscopy improves the sensitivity of the examination from 60% to 90%, compared to visual inspection of the lesion [18]. Various algorithms have been developed to increase the sensitivity and specificity of dermoscopy—ABCDE rule, pattern analysis, 7-point checklist, and the Menzies method [17,18]. The application of these algorithms allows for achieving sensitivity and specificity above 90% [18]. However, the accuracy of dermoscopy strictly depends on the experience of the examinator, achieving similar effectiveness for inexperienced doctors as visual inspection with an unaided eye [18,19]. Reflectance confocal microscopy (RCM) is another non-invasive method of imaging skin lesions with magnification on a cellular level [20]. This technique uses a near-infrared laser to image individual skin layers up to the superficial dermis. It can be used to diagnose skin lesions, monitor the efficiency of non-invasive therapies, and assess tumor recurrence after surgical excision. The disadvantages of this method are the long time of image acquisition, difficulty in imaging lesions on curved surfaces, and the need for special equipment and specialized training to evaluate images properly [21,22]. The sensitivity and specificity of the RCM are estimated at 92% [22]. Another non-invasive method of diagnosing pigmentary lesions is optical coherence tomography (OCT). This technique uses the low-coherence infrared laser light to produce two- or three-dimensional images of examined tissue by detecting scattered light. It offers higher penetration into the skin layers but at the cost of lower resolution compared to RCM. It can also be less effective in imaging pigmented lesions due to melanin light absorption [23,24]. As with RCM, it also requires specialized equipment and training to interpret images correctly. OTC sensitivity for melanoma diagnosis varies between 79–94%, and its specificity is between 85–96% [23]. Ultra-high frequency ultrasonography is also applied in the imaging of pigmented lesions. Initially, it was utilized to measure the thickness of the infiltration of the lesion into the skin. Ultrasound waves of frequency higher than 20 MHz can be applied for imaging various skin pathologies, but insufficient resolution limits its use in the differentiation of benign lesions and melanomas. To some extent, the potential for differentiating pigmented lesions may be increased by applying color Doppler ultrasound [25,26] This method offers sensitivity and specificity at 83% and 74%, respectively [26]. Another non-invasive technique for the diagnosis of pigmented lesions is tape stripping. In this method, adhesive tape is used to collect the thin layer of stratum corneum cells for further analysis [27]. Combined with Gene Expression Profiling (GEP) in the case of melanoma this technique allows to predict the tumor progression and risk of metastases and to personalize the treatment. It also allows for distinguishing melanomas from benign nevi with sensitivity and specificity above 90% [28].

A promising direction in developing differential diagnosis of pigmentary lesions is automatic computer-aided diagnostic systems. These tools, independent of operator skills, may revolutionize the diagnosis of skin lesions in the future. They are based on an objective and measurable analysis of individual elements of the image of a pigmented lesion obtained with one of the selected imaging methods. Artificial intelligence and machine learning have the potential to achieve much higher sensitivity and specificity than a clinical examination, even if performed by an experienced clinician. However, the currently available systems show sensitivity and specificity at the level of 90.1% and 74.3%, respectively [29]. This excludes them as the only test in the diagnosis of pigmented lesions, and they should only supplement the clinical examination. However, due to the potential of this type of diagnostics, it is recommended to further improve it by searching for new, measurable image parameters usable in computer analysis. One of such parameters may be the fractal analysis of shape and texture.

In classical, Euclidian geometry, we are used to thinking that dimension is an integer value. For instance: the 0 is a dimension of a point, a section has one dimension—length, whereas length and width describe plane figures, and solids have three dimensions: length, width, and height. Fractals go beyond those simple rules. Their dimensions are rational numbers and may take values between 0 and 3. Self-similarity is another feature of fractals. It seems that on any scale of observation, fractals look similar. Benoît Mandelbrot is the father of fractals. He described the principles of fractal geometry in 1982 [30]. It is easy to describe the shapes of simple figures. Nature, especially living forms, is full of shapes that may be treated as fractals up to a certain scale, such as neuron networks and nets of blood vessels. It is hard to describe these shapes using simple Euclidian geometry. It is possible to measure area or perimeter, but a more complex shape needs to be detailed, and it is harder to do that. Calculation of fractal dimension (FD) enables obtaining a fractional number that describes the examined shape. It is a relation between complexity and the examined shape. The more complex it is, the value of FD is lower. An example of this relationship is shown in Figure 1.

The fractal dimension analysis is widely used in medicine to analyze the irregularities of the examined CT, MRI, and USG images, even for chromatin measurements of melanoma cells [31,32,33,34].

This study aims to apply fractal dimension analysis to differentiate melanomas, dysplastic nevi, and benign nevi in polarized and non-polarized light. We put the following null hypotheses:There are no differences in area, perimeter, and fractal dimensions of lesions in normal and polarized illumination in each examined group (melanoma, dysplastic nevus, benign nevus);There are no differences in fractal dimensions of lesion shapes in the groups (melanoma, dysplastic nevus, benign nevus);There are no differences in fractal dimensions of lesion surface structures in the groups (melanoma, dysplastic nevus, benign nevus);There is no correlation between fractal dimensions and the area or perimeter of lesions in normal and polarized light in the groups (melanoma, dysplastic nevus, benign nevus).

## 2. Materials and Methods

### 2.1. Patients and Lesions

A total of 102 pigmented lesions of 88 patients of Wroclaw Comprehensive Cancer Center were included in this study. All lesions were examined by a dermatologist and, as potentially malignant, referred to the Skin Cancer Unit of Wroclaw Comprehensive Cancer Center. Before surgical excision, they underwent secondary examination and were qualified for the procedure by a surgical oncologist (M.Z.), according to the Skin Cancer Unit protocol. All lesions were photographed under polarized and non-polarized light before the surgery. Excision was performed with 1% lidocaine as a local anesthetic agent. All lesions were examined by the histopathologist experienced in diagnosing skin lesions. After histopathological examination, all lesions were divided into three groups: benign nevi (BN), dysplastic nevi (DN), and melanoma (MM). The BN group contained only intradermal nevi, junctional nevi, and compound nevi. Other types of benign lesions, due to a different clinical picture that could influence the results, were excluded from the study. DN group included lesions meeting the criteria described in the dedicated paragraph of the introduction. The MM group included the superficial spreading melanomas and lentigo maligna melanomas as these types can be most easily confused with benign lesions. Nodular melanomas, due to their three-dimensional structure, which could distort the results of the analysis, were also excluded from the study. Pigmented lesions located on the hairy scalp or oral mucosa and lesions with erosion or ulceration were excluded from the study. For further analysis, we enrolled 84 patients with 97 lesions (38 men and 46 women). The mean age of patients was 53 years, and the median was 50 years, ranging from 14 to 90 years. Informed consent was obtained from all patients. Among the examined lesions, histopathological examination revealed 20 melanomas (8 in situ and 12 invasive), 23 dysplastic nevi, and 54 benign nevi. The average depth of invasive melanomas was 1.01 mm (SD = 0.69) with average 2.1 mitoses (SD = 2.2). The project of the study was approved by the Bioethics Committee of the Wroclaw Medical University, approval number KB—502/2019 (27 May 2019).

### 2.2. Photographs

All photos were taken using a Canon EOS 77D (Canon, Ōta, Tokyo, Japan) camera with a Dermlite Foto II Pro lens (3Gen Inc., San Juan Capistrano, CA, USA). The resolution of the images was 6000 × 4000 pixels with sRGB color space. All images have the same parameters: ISO 800, time of exposition 1/60 of a second. All lesions were photographed in the same exposition under non-polarized and polarized illumination. The applied lens enables shooting repeatable images on the same scale and the stable lighting condition, obtaining comparable photos of different lesion types. It also offers a built-in millimeter-scale enabling precise measuring of the lesion.

### 2.3. Basic Image Analysis

The area and perimeter of all lesions were measured. Both non-polarized and polarized light photographs of lesions were analyzed. All measurements were made using ImageJ software, version 1.53e (Image Processing and Analysis in Java—Wayne Rasband and contributors, National Institutes of Health, Bethesda, MD, USA, public domain license, https://imagej.nih.gov/ij/, accessed on 1 January 2022). ImageJ enables measuring area and perimeter after drawing the lesion contour. This operation was carried out by two independent researchers (P.P. and K.J.) on color images before any graphical modifications. All measurements were performed separately for the image of each lesion in polarized and non-polarized light.

### 2.4. Image Processing

#### 2.4.1. Preparing Images for Shape Analysis

All graphic operations were performed in GIMP version 2.10.30 (GNU Image Manipulation Program—www.gimp.org (accessed on 2 February 2022), free and open-source license). Figure 2 shows operations that lead to the segmentation of examined lesions. The first color image was converted into 8-bit grayscale. The tones of grayscale photographs were inverted. The crucial operation was the division into blending grayscale and inverted images. The grayscale image was set as a bottom layer, and the inverted grey scaled image was set as an upper layer. In this operation, each pixel of the lower layer was multiplied by 256. One was added to the value of each pixel of the upper layer (to prevent dividing by zero). The value of the lower-layer pixel was divided by the value of the corresponding upper-layer pixel. This operation is described by the following formula:(1)$$E_{x,y}=\frac{I_{x,y}\times 256}{M_{x,y}+1}.$$
where: *E*—final pixel value, *x*,*y*—coordinates of each pixel of an image, *I*—pixel value of the lower layer image, *M*—pixel value of the upper layer image.

This operation led to achieving a separated lesion on a white background ready for further shape analysis. This operation was enough, in most cases. If still some artifacts in the backgrounds remained, they were manually corrected (in a few cases, this algorithm gave some dots and stains in the background. In these cases, these artifacts in the backgrounds were manually erased from the background). The main boundary criterion of this operation was to be selected as near to the lesion as possible, without containing any melanin spots. Such prepared images underwent further fractal dimension analysis in the aspect of lesion shape.

#### 2.4.2. Preparing Images for Surface Feature Analysis

We extracted multiple regions of interest (ROIs) from each lesion image to represent the most characteristic areas within the lesion borders. The smallest lesion size determined the ROI size, set to 450 × 450 pixels. We extracted multiple ROIs from larger lesions, without overlapping, so that they represent the entire lesion area, not just the section selected by the researcher. Thanks to this, we avoided the bias consisting of the deliberate selection of areas with a regular structure from benign lesions and the most irregular areas from malignant lesions, which could affect the result of the analysis. ROIs were taken from all lesions in normal and polarized illumination. The region of interest covers only the lesion surface, without surrendering skin. Therefore, the number of ROIs depends on lesion size. The larger the lesion area, the greater number of ROIs. For further fractal dimension analysis of the surface, we extracted 200 ROIs from the melanoma group, 107 from dysplastic nevi, and 198 from benign nevi for polarized and non-polarized light images. Each ROI was converted into an 8-bit grayscale image.

### 2.5. Fractal Dimension Analysis

The ImageJ version 1.53e (Image Processing and Analysis in Java—Wayne Rasband and contributors, National Institutes of Health, Bethesda, MD, USA, public domain license, https://imagej.nih.gov/ij/, accessed on 1 January 2022) and the FracLac plugin version 2.5 (Charles Sturt University, Bathurst, Australia, public domain license) were used to perform all fractal analyses.

We applied the modified algorithm to the classical counting box method—the intensity difference fractal dimension counting method. This algorithm allowed us to analyze 8-bit monochromatic images. It was fully described in our previous study [35]. In this method, the analyzed image is divided into boxes like compartments in the classical counting box method. The difference between the maximum and minimum pixel intensity is counted in each box:δI_i,j,ε_ = maximum pixel intensity i,j,ε − minimum pixel intensity i,j,ε(2)
where: δI—the difference between maximum pixel intensity and minimum pixel intensity i,j—coordinates of the analyzed box in a scale ε.

In the next step, 1 is added to the intensity difference to avoid its value being 0:I_i,j,ε_ = δI_i,j,ε_ + 1(3)

Finally, the fractal dimension of the intensity difference is described by the following formula:(4)$$\mathrm{FD}=\underset{\varepsilon \to 0}{\lim}\left(\frac{\ln I\left(\varepsilon \right)}{\frac{1}{\varepsilon }}\right)$$
where: FD—fractal dimension of intensity difference, I(ε) = Σ [δIi,j,ε + 1], ε—scale of box.

Two different methods of scale ε calculation were applied. For lesion shape analysis, power series was set with base 2 and exponent 2 (for example, first three dimensions of ε are: 2^2^ = 4, 2^(2+2)^ = 16, 2^(4+2)^ = 64). In the case of lesion surface analysis, block series was applied. This option scans a square block within an image using a series of grids calculated from the block size. This feature is most useful for analyzing patterns that fill the whole area of an image.

### 2.6. Statistical Analysis

Statistica version 13.3 (StatSoft, Cracow, Poland) was applied to calculate all statistical tests, and 0.05 was set as the statistically significant level. The normality of distribution was confirmed by the Shapiro–Wilk test. Due to normal distribution, we performed parametric tests. We applied the paired t-Student test (check for differences between lesion area, perimeter, fractal dimension in normal and polarized light). Analysis of variance (ANOVA) and post hoc least significant difference were applied to show differences in fractal dimensions between lesions in normal and polarized illumination. In case of lack of normal distribution, we turned to non-parametric tests: (Kruskal-Wallis, Wilcoxon, Mann–Whitney U tests). The correlation matrix was applied to calculate the Spearman correlation coefficient^®^) between fractal dimensions of lesions in various illumination and area and perimeter. Normal distribution was seen in the case of fractal dimension of shape. Lack of normal distribution was seen in the case of the perimeter (and associated values for example area, etc.) and surface fractal dimension.

## 3. Results

The age structure, age of subjects, and location of pigmented lesions are shown in Table 1. In the group of melanomas, there were more men, while benign and dysplastic lesions were more often observed in women. The most common localization of examined lesions was the trunk area.

The results of measurements analysis of the perimeter and the area of pigmented lesions in polarized and non-polarized light are presented in Table 2. It is noteworthy that the measurement values were higher for polarized light across all subgroups. This difference was statistically significant for the perimeter within all lesions and the surface area of benign and dysplastic lesions.

A comparison of the area and perimeter of benign nevi, dysplastic nevi, and melanomas is presented in Table 3. The melanoma subgroup showed significantly higher values in all parameters in relation to both benign and dysplastic nevi. The difference in the parameters between benign and dysplastic lesions did not show statistical significance.

Table 4 shows the results of a fractal dimension analysis of the shape of benign, dysplastic nevi and melanomas and compares their values between polarized and non-polarized light. In all groups, values of FD were significantly higher in non-polarized light (*p* < 0.001). Sample images of pigmented lesions representing individual groups along with their fractal dimensions of shape values in polarized and non-polarized light are shown in Figure 3.

A comparison of fractal dimensions of lesion shapes of benign nevi, dysplastic nevi, and melanomas is shown in Table 5. The fractal dimension of the lesion shape distinguishes melanomas very well from benign and dysplastic nevi and dysplastic nevi from benign nevi under polarized light. It should be noted that in the case of non-polarized light, the fractal dimension of shape allows distinguishing benign nevi from dysplastic nevi and melanomas. Still, there are no significant differences between dysplastic nevi and melanomas.

The Spearman correlation coefficient between the value of the fractal dimension of shape (FD) and the area and perimeter showed a strong negative correlation for benign lesions under polarized light and a moderate negative correlation under non-polarized light for all analyzed parameters. The results of this analysis are shown in Table 6.

The results of fractal dimension for lesion surface (ROIs) showed no statistical differences within particular groups between polarized and non-polarized light. Still, the comparison between the subgroups demonstrated that the fractal dimension for surface distinguishes very well the surface of melanoma from dysplastic and benign nevi under polarized light. However, this analysis shows no difference between the particular lesions in non-polarized light and cannot be used to distinguish benign from dysplastic nevi under polarized light. The result of the surface fractal dimension for particular lesions is shown in Table 7, and the comparison between subgroups is shown in Table 8.

We also conducted a separate analysis in the group of melanomas to illustrate the differences between in-situ melanomas and invasive melanomas. Both the fractal dimension of the shape and the fractal dimension of the surface showed statistically significant differences, regardless of whether the analysis concerned polarized or non-polarized light. The results are shown in Table 9 and Table 10.

## 4. Discussion

The age and sex structure for melanomas, benign and dysplastic nevi correspond to the data reported by other authors [36,37]. The higher age of melanoma patients is due to the cumulative effect of risk factors throughout life, particularly ultraviolet (UV) exposure. UV protection measures are used less frequently by men, who are more commonly affected by melanoma [38,39]. A statistically higher value of the surface area and perimeter of melanomas in relation to benign and dysplastic changes is also observed in the other studies [40]. Irregularity of the border in melanoma leads to a substantial extension of the perimeter of the lesion. It was significantly higher for melanomas than benign and dysplastic nevi, which results from both the grater area of the lesion and greater irregularity of the lesion boundaries in the case of a malignant lesion. These results are also reflected in the commonly used algorithms of the clinical examination of pigmented lesions—B—border and D—diameter of ABCDE rule and border irregularity and large diameter in 7-point checklist [14,41]. Our work confirms the value of these features in the differential diagnosis of melanoma. Another observation is the tendency to underestimate the dimensions of the change when assessing it in non-polarized light in relation to polarized light. Because of reduced light reflection, polarized light provides higher contrast between the lesion and the surrounding skin. This difference was statistically significant for all the measured parameters for all types of the examined lesions, except for the surface area of melanomas, where the difference was not statistically significant due to the relatively large dimensions of lesions. This finding could have an important impact on clinical practice. A surgeon who assesses the lesion only by visual inspection in non-polarized light, usually also without magnification, may incorrectly assess the extent of the lesion borders, which may lead to incomplete resection during surgery. Indeed, data from the literature show that positive margins are present in 4.2–46% of the surgical excisions of pigmented lesions [42,43]. The presence of melanocytes in the margin of the histopathological specimen is not clinically significant in the case of benign and dysplastic lesions. Kim et al. observed no melanoma occurrence at the site of 467 previously excised dysplastic nevi with positive margins in histopathological examination with a mean 6.9 years follow-up. Recurrence in the pigmentation at the site was very low and occurred in 1.2% of the cases [44]. In contrast, positive or very narrow margins in the case of melanoma may worsen the patient’s overall prognosis [45,46]. This result suggests that not only can dermatologists benefit from dermoscopy performed under polarized light in terms of accurate diagnosis, but it can also be beneficial for surgeons with regard to the proper determination of the extent of the lesion and planning its surgical excision. The superiority of polarized light over non-polarized light in the assessment of melanocytic changes is also indicated by the results of the fractal dimension analysis. Fractal dimension enables analyzing shape or surface details. The lowest value of FD is presented in the lowest regularity of the analyzed shape [47,48,49]. For each of the studied groups, the fractal dimension values were significantly lower in polarized light compared to non-polarized light. It is measurable proof that polarized light shows the irregularity of the structures of pigmentary lesions significantly better than non-polarized light. This indicates the high value of polarized light in the differential diagnosis of pigmented lesions. This observation confirms and additionally strengthens the observations of other authors [15,50,51]. Comparison of fractal dimension of shape in the examined groups proved its great value in differentiating melanomas from benign and dysplastic lesions and benign from dysplastic nevi under polarized light. The lower values of the fractal dimension of shape are associated with the increasing degree of cytological atypia of the lesions, reflecting the irregularity of examined structures. The numerical result of this analysis for a given lesion shows it can also be used to monitor the evolution of a nevus under clinical observation. Under non-polarized light, only benign lesions can be distinguished from dysplastic nevi and melanomas, additionally emphasizing the superiority of polarized light. Only a few studies using the fractal dimension in assessing pigmented lesions were found. Piantanelli et al. presented an analysis of pigmented lesion boundary, finding it useful in differentiating melanomas both from benign and dysplastic lesions [52]. Carbonetto et al. successfully used the fractal dimension of the lesion border for developing an automatic system for lesion classification into melanomas and benign lesions, but it was applied to a very limited number of cases [53]. On the other hand, Cross et al. showed no significant differences in the fractal dimension of melanomas and benign melanocytic lesions [54]. It may be due to a very small number of compared cases, low quality of photographs of the examined lesions, and a different technique of determining the fractal dimension. We observed a tendency of a growing correlation coefficient between FD of shape in polarized and non-polarized light and Euclidian geometry parameters (area and perimeter). Lack of or weak correlation was noted in melanomas, moderate in the DN group, and the highest value of the r parameter was observed in the BN group. It is interesting that in the MM (N-PL) group, a weak positive correlation was noted between the fractal dimension of shape and Euclidian parameters in contrast to DN and BN groups where r values suggest a weak negative (DN) and moderate negative (BN) correlation. It may be explained by the greatest standard deviation of lesion area in the MM group in contrast to the same lowest feature in the BN group. In our previous study, we confirmed a moderate negative correlation between FD of the surface lesion in the case of the potentially malignant lesion (oral lichen planus). In that study, we observed that the surface FD of lesions was significantly lower than normal mucous [55]. The analysis of the fractal dimension of the surface of the examined lesions turned out to be a less differentiating parameter than the fractal analysis of the lesion shape. Although the mean value of the fractal dimension was lower in the case of polarized light, which indicates a greater detail of the imaged structures, this difference was not statistically significant compared to non-polarized light across all groups. The results of individual groups correlate with the increasing degree of atypia of lesions, assuming the lowest values for melanomas, which corresponds to the highest degree of irregularity in the structure of the disease. This measurement enables the differentiation of benign and dysplastic lesions from melanomas under polarized light but not between benign and dysplastic lesions. Analysis in non-polarized light shows a complete lack of differences between individual groups, which allows concluding that this index is not useful for differentiating lesions in non-polarized light. This descriptor proved to be less useful than the fractal dimension of the shape in our study. It is focused on the measurement of pigment chaotic distribution within the lesion borders. Perhaps atypical structures of melanomas and dysplastic lesions may form some larger homogenous areas within the lesion surface, which weaken the importance of this parameter. Perhaps the lack of statistical differences was also due to the extraction of many ROIs from larger lesions, representing both more and less irregular melanin distributions within particular lesions. The lack of differences may also result from the limited number of compared lesions. Despite the lower diagnostic value of the analysis of the fractal dimension of the surface in relation to the fractal dimension of the shape, it was applied successfully by Moldovanu et al. to develop an automatic skin lesion classification system [56]. They managed to achieve 95% accuracy in differentiating melanomas from nevi using Higuchi’s surface fractal dimension combined with color features. This excellent result shows the potential of this parameter, but in relation to the methodology of this study and our results, this parameter should be used as part of multivariate image analysis, not as an only differentiating parameter. It should also be noted that this analysis requires photographs with very great detail and should only be used to compare images with the same exposure parameters at the same magnification.

The results of the comparison of the fractal dimension values for invasive and non-invasive melanomas clearly indicate that both the fractal dimensions of shape and surface can be used to differentiate these changes successfully. Knowing the nature of the lesion before the final result of the histopathological examination of excised tissue is obtained, can have an impact on more effective planning and faster introduction of the treatment process [57,58]. However, the results of this comparison may be reliable only up to a limited extent due to the small size of the compared groups.

## 5. Conclusions

Fractal dimension analysis of images in polarized light enables distinguishing melanomas, dysplastic nevi, and benign nevi in shape. It also makes it possible to distinguish melanomas from benign and dysplastic nevi under non-polarized light.Fractal dimension of texture allows distinguishing melanomas from benign and dysplastic nevi under polarized lightPolarized light is superior to non-polarized light for imaging details of pigmented lesions, especially their borders. It offers better material as the base for fractal dimension analysis than non-polarized light images.It is possible to distinguish between in situ and invasive melanoma in shape and surface structure (in both polarized and non-polarized light).Fractal dimensions of shape and texture are parameters useful for developing automatic computer-based diagnosis systems.There is no correlation between the fractal dimensions and area or perimeter of lesions in normal and polarized light in the melanoma and dysplastic nevus groups.There is a negative correlation between the fractal dimensions and area or perimeter of lesions in normal (r = −0.45) and polarized light (r = −0.72) in the benign nevus group.

In our study we have rejected the following null hypotheses:There are no differences in area, perimeter, and fractal dimensions of lesions in normal and polarized illumination in the examined groupThere are no differences in fractal dimensions of lesion shapes in the groups (melanoma, dysplastic nevus, benign nevus);There are no differences in fractal dimensions of lesion surface structures in the groupsThere is no correlation between the fractal dimensions and area or perimeter of lesions in normal and polarized light in the benign nevus group.

We have also confirmed the null hypothesis that there is no correlation between the fractal dimensions and area or perimeter of lesions in normal and polarized light in the melanoma and dysplastic nevus group.

## 6. Study Limitations

The results of the proposed analyses may be disturbed by a large number of hairs in the preparation or ulcers and other damage within the lesion; therefore, it should not be performed on lesions located on the hairy scalp and lesions with erosion or ulceration. The comparison also requires high-quality photos. In the fractal dimension of surface analysis, they must be made with the same equipment, magnification, and stable exposure conditions. Comparison between melanoma in situ and invasive may not be reliable due to a limited number of compared lesions and requires further research.

## Figures and Tables

**Figure 1 life-12-01008-f001:**
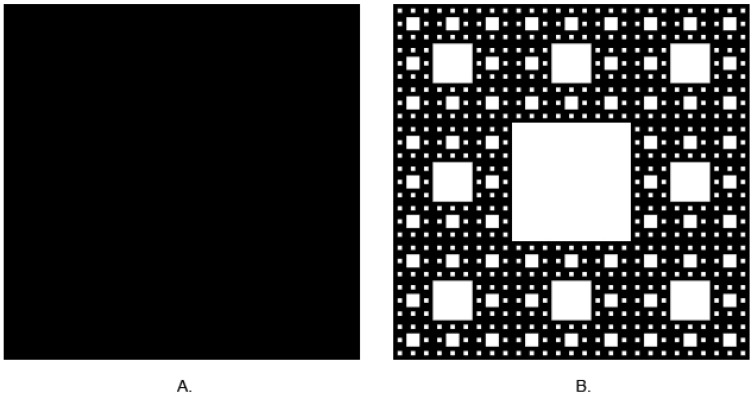
(**A**) Square (FD = 2), (**B**) Sierpinski carpet (fFD ≈ 1.8928.) (Generated by https://codinglab.huostravelblog.com/math/fractal-generator/ accessed on 1 January 2022).

**Figure 2 life-12-01008-f002:**
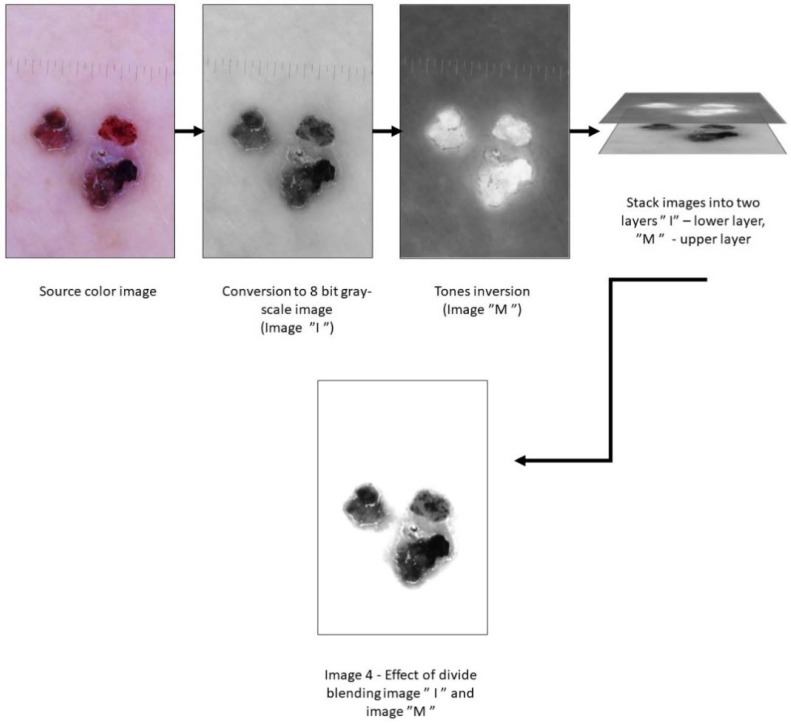
Preparing images for shape analysis.

**Figure 3 life-12-01008-f003:**
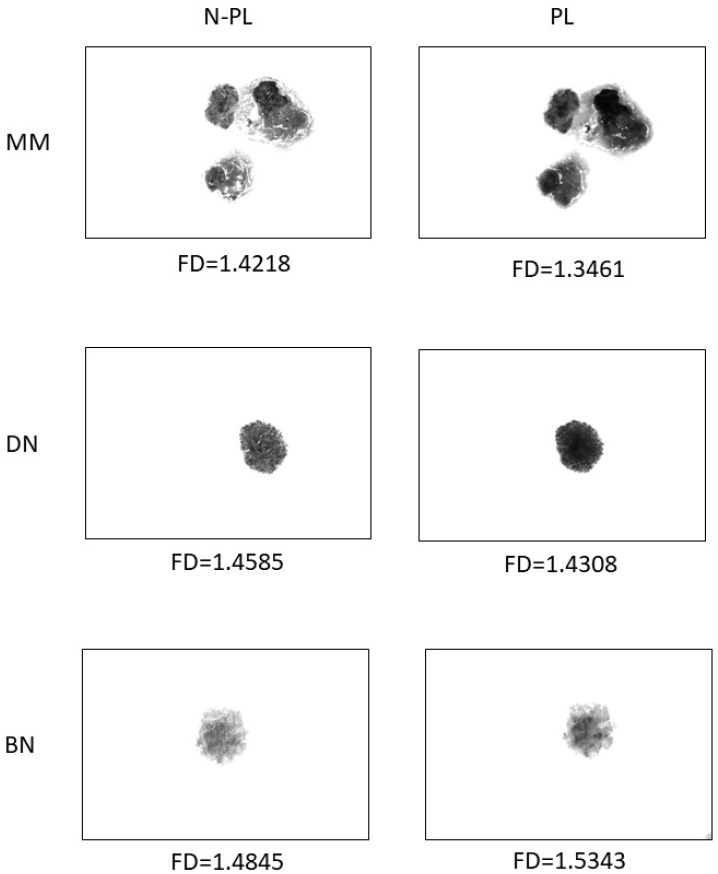
Images of pigmented lesions representing individual groups along with their fractal dimensions of shape values in polarized (PL) and non-polarized (N-PL) light.

**Table 1 life-12-01008-t001:** Structure of sex and age of subjects and localization of examined lesions.

Lesion	Sex		Age				Localization				
	F	M	Mean	SD	Min.	Max.	Head/Neck	Trunk	Upper Limb	Lower Limb	Total
MM	8	12	62.0	18.3	33	90	1	9	6	4	20
DN	10	8	55.6	18.1	23	81	2	17	2	2	23
BN	28	18	46.0	18.4	14	85	7	34	2	11	54
Total	46	38	54.6	18.3	14	90	10	60	10	17	97

**Table 2 life-12-01008-t002:** Results of the Wilcoxon test for examined Euclidian features (MM—melanoma, DN—dysplastic nevus, BN—benign nevus, SD—standard deviation, PL—polarized light, N-PL—non-polarized light, N—number of lesions, *p*—*p*-value).

Lesion	Examined Feature	Mean	SD	N	Difference	*p*
MM	Area of PL [mm^2^]	166.95	210.16	20	5.39	0.4781
	Area of N-PL [mm^2^]	161.57	206.95			
DN	Area of PL [mm^2^]	18.43	12.02	23	0.92	0.0081
	Area of N-PL [mm^2^]	17.51	11.39			
BN	Area of PL [mm^2^]	15.51	13.95	54	0.61	0.0000
	Area of N-PL [mm^2^]	14.91	13.66			
MM	Perimeter of PL [mm]	54.73	33.34	20	4.16	0.0012
	Perimeter of N-PL [mm]	50.57	30.61			
DN	Perimeter of PL [mm]	18.55	7.81	23	1.11	0.0012
	Perimeter of N-PL [mm]	17.44	6.91			
BN	Perimeter of PL [mm]	14.56	7.29	54	0.67	0.0000
	Perimeter of N-PL [mm]	13.89	6.74			

**Table 3 life-12-01008-t003:** Results of the Kruskal-Wallis test for the multiple comparison of area and perimeter of lesions in polarized and non-polarized light (MM—melanoma, DN—dysplastic nevus, BN—benign nevus, R—mean rank, *p*—*p* value).

Polarized Light				Non-Polarized Light			
Area				Area			
	MM R = 80.700	DN R = 47.739	BN R = 37.796		MM R = 81.450	DN R = 47.652	BN R = 37.565
MM		*p* = 0.000384	*p* = 0.000000	MM		*p* = 0.000257	*p* = 0.000000
DN	*p* = 0.000384		*p* = 0.467883	DN	*p* = 0.000257		*p* = 0.448982
BN	*p* = 0.000000	*p* = 0.467883		BN	*p* = 0.000000	*p* = 0.448982	
Perimeter				Perimeter			
	MM R = 84.000	DN R = 48.870	BN R = 36.093		MM R = 84.350	DN R = 48.565	BN R = 36.093
MM		*p* = 0.000134	*p* = 0.000000	MM		*p* = 0.000096	*p* = 0.000000
DN	*p* = 0.000134		*p* = 0.204818	DN	*p* = 0.000096		*p* = 0.225339
BN	*p* = 0.000000	*p* = 0.204818		BN	*p* = 0.000000	*p* = 0.225339	

**Table 4 life-12-01008-t004:** Results of the paired t-Student for fractal dimension of shape in polarized and non-polarized light (MM—melanoma, DN—dysplastic nevus, BN—benign nevus, SD—standard deviation, PL—polarized light, N-PL—non-polarized light, FD—fractal dimension of shape, N—number of lesions, *p*—*p*-value).

Lesion	Examined Feature	Mean	SD	N	Difference	*P*
MM	FD PL	1.3885	0.0404	20	−0.0586	0.0000
	FD N-PL	1.4471	0.0431			
DN	FD PL	1.4225	0.0393	23	−0.0459	0.0000
	FD N-PL	1.4685	0.0318			
BN	FD PL	1.4713	0.0579	54	−0.0255	0.0001
	FD N-PL	1.4968	0.0418			

**Table 5 life-12-01008-t005:** Post hoc ANOVA results (least significant difference) for comparing shape FD values of lesions in polarized and non-polarized light (MM—melanoma, DN—dysplastic nevus, BN—benign nevus).

Polarized Light				Non-Polarized Light			
Value of Fractal Dimension for the Shape of Lesions							
	MM	DN	BN		MM	DN	BN
MM		0.031161	0.000000	MM		0.144874	0.000165
DN	0.031161		0.000275	DN	0.144874		0.021539
BN	0.000000	0.000275		BN	0.000165	0.021539	

**Table 6 life-12-01008-t006:** The Spearman correlation coefficient between the value of fractal dimension of shape (FD) and the area and perimeter in polarized (PL) and non-polarized light (N-PL). (MM—melanoma, DN—dysplastic nevus, BN—benign nevus).

	MM		DN		BN	
	PL	N-PL	PL	N-PL	PL	N-PL
FD vs. Area of lesion	−0.131579	0.303759	−0.367589	−0.219368	−0.720775	−0.453937
FD vs. Perimeter of lesion	0.001754	0.321805	−0.213439	−0.265810	−0.712869	−0.459186

**Table 7 life-12-01008-t007:** Mean and standard deviation (SD) of fractal dimension surface (ROIs) of lesions in polarized and non-polarized light.

Lesion	Polarized Light		Non-Polarized Light	
	Mean	SD	Mean	SD
MM N = 200	1.4117	0.1639	1.5909	0.1160
DN N = 107	1.4498	0.1314	1.6139	0.0880
BN N = 198	1.5007	0.1314	1.5965	0.0930

**Table 8 life-12-01008-t008:** Results of Kruskal–Wallis test for the multiple comparisons of surface fractal dimension of lesions in polarized and non-polarized light (MM—melanoma, DN—dysplastic nevus, BN—benign nevus, R—mean rank, *p*—*p* value).

Polarized Light				Non-Polarized Light			
Fractal Dimension Values for Lesion Surface (ROIs)							
	MM R = 300.07	DN R = 241.16	BN R = 211.42		MM R = 246.04	DN R = 275.08	BN R = 256.10
MM		*p* = 0.002331	*p* = 0.000000	MM		*p* = 0.300678	*p* = 1.000000
DN	*p* = 0.002331		*p* = 0.267457	DN	*p* = 300678		*p* = 0.854320
BN	*p* = 0.000000	*p* = 0.267457		BN	*p* = 1.000000	*p* = 0.854320	

**Table 9 life-12-01008-t009:** Results of the unpaired t-Student test—differences of shape fractal dimension between MM in situ and MM invasive (MM—melanoma, SD—standard deviation, PL—polarized light, N-PL—non-polarized light, FD—fractal dimension of shape, N—number of lesions, *p*—*p*-value).

Examined Feature	Lesion	Mean	SD	N	*p*
FD PL	MM in situ	1.4166	0.0165	8	0.0165
	MM invasive	1.3708	0.0165	12	
FD N-PL	MM in situ	1.4800	0.0173	8	0.0173
	MM invasive	1.4300	0.0173	12	

**Table 10 life-12-01008-t010:** Results of the Mann–Whitney U test—differences of surface fractal dimension between MM in situ and MM invasive (MM—melanoma, SD—standard deviation, PL—polarized light, NL—natural light, FD—fractal dimension of surface, N—number of ROIs, *p*—*p*-value).

Examined Feature	Lesion	Mean	SD	N	*p*
FD PL	MM in situ	1.4676	0.1367	86	0.0059
	MM invasive	1.3678	0.1500	114	
FD N-PL	MM in situ	1.6241	0.0877	86	0.0065
	MM invasive	1.5639	0.1557	114	

## Data Availability

Data are available at pawel.popecki@umw.edu.pl (P.P.) and kamil.jurczyszyn@umw.edu.pl (K.J.).

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
