# Peer review of "Fractal Dimension Analysis of Melanocytic Nevi and Melanomas in Normal and Polarized Light—A Preliminary Report"

_life, 2022, doi:10.3390/life12071008_

Round 1

Reviewer 1 Report

Resume

The thematic of the paper is relevant and corresponds to the Journal. Studies theme  is very relevant. This is very well described in the introduction. There are significant comments on the methodology and data statistical analysis. The paper has inaccurate it is need to correct them carefully.

Comments

Замечания:

  1. In the introduction, information should be added about the features of dermatoscopy and dermatoscopic signs of pigmented skin lesions. Then it will become clear why fractal analysis can solve the problem of morphometry of the texture of dermatoscopic images.
  2. In line 164, 167-168: “20+23+54” not equal to “98”. Lost 1 lesion.
  3. In line 198-204: In the text, formula and Figure 2, the descriptions of images M and I differ: it is not clear where the image M is, and where I is?
  4. 2.4.1. Preparing Images for Shape Analysis:

How correct is the described image segmentation? Such image segmentation is equivalent to threshold processing with a threshold of 127: taking into account the normalization of values exceeding 255. The authors describe the subsequent manual correction of segmentation results (not in detail). In general, all this leads to a loss of information about the true boundary of the lesion, which significantly affects the results. At the same time, the authors refer to articles 49, 50, 51, 53, which show the high importance of image segmentation. Article 49 has a clear criterion for the correct segmentation of images: pixel of bounds “to be selected as near to the lesion as possible, and it must not contain melanin spots”. This image segmentation will be the reference. The most accurate method of automatic image segmentation is chosen by comparing with this reference method. Otherwise, the images are segmented manually.

  1. 2.4.2. Preparing Images for Surface Feature Analysis:

2.4.2. is not clear. Regions of interest extracted with overlap? The smallest lesion corresponds to one 1 ROI? Extracting multiple ROIs from a single lesion is not correct in statistical data analysis.

  1. In line 255: "Diameter" is an error, because earlier and later in the article "perimeter" is used.
  2. Tab 1. SD is the same for all: this is hardly possible.
  3. In line 268: “The results of measurements” - more  rigorous “The results of measurements analysis”
  4. In line 271: “Circumference” is an error, because earlier and later in the article "perimeter" is used.
  5. In line 269-272 That's how it should be. “Polarized dermoscopy reduces the possibility of reflection, which leads to visualizing even deeper layers of lesion than in non -polarized  light [15].” This leads to different contrast and therefore to different image segmentation results.
  6. Tab. 2: What is “Difference” and “SD Difference”? How to calculate them? Perhaps “Difference” is the difference of the mean values (if calculated according to the table data). It is not clear how to calculate “SD Difference”?
  7. Tab. 3: The data in the table resembles correlation matrices! It doesn't look like a result of the Kruskal-Wallis test. Test result is rank, test statistic and significance value.
  8. In line 283-287: see 10.
  9. Tab. 4: see 11.
  10. Tab. 5: Need to clarify the method of Post hoc ANOVA its parameters.
  11. Tab. 7: The table is not informative. It is necessary to compare the data statistically.
  12. Tab. 7: see 12.
  13. In line 349-358: see 10.
  14. In line 406-408: It is strange that the form of lesions turned out to be more informative. In dermatoscopy of melanocytic lesions, texture patterns are most significant: the presence or absence of a pigment network, atypical stripes, dots, lacunae, atypical vascular structure, and so on.
  15. Statistical Analysis generally: It is not described which parameters had a normal distribution, which did not. Small lesions are more common, large ones are less common. Therefore, parameters such as area and perimeter associated with dimensions will not have a normal distribution.
  16. Null hypotheses are formally described, but formally there are no final conclusions.
  17. Wherever it is written fractal dimension, it is necessary to write that this is the fractal dimension of the shape or texture, so that there is no confusion.

Correcting these comments will significantly improve the article.

Reviewer 2 Report

Thank you for sharing this analysis “Fractal Dimension Analysis of Melanocytic Nevi and Melanomas in Normal and Polarized Light – A Preliminary Report.“

For introduction it is appropriate to mention noninvasive diagnostics such as GEP, tape stripping which have promising data for adult pigmented lesions.

It would be helpful to provide sample sensitivity/specificity for the approaches introduced, for comparison.

Please revise for clarity, am not sure what you’re saying: “Artificial intelligence and machine learning may allow in the future to reach specificity and sensitivity of the test not available for a clinical examination.”

The authors have acknowledged that they have already excluded a number of pigmented lesions that might be challenging to analyze - presumably this would include Spitz nevi and atypical Spitz tumors, nodular melanomas, etc.  There is often diagnostic discrepancy between severely dysplastic nevi and melanoma in situ — please provide a comparison of these two diagnoses specifically and whether they can be distinguished.  Also please provide overall information for melanomas (average, st dev for depth, mitoses, ulceration, etc.)

In discussing novelty and exciting results, the paper is written in a way that suggests that most dermatologists are not using dermoscopy with polarized light (which I don’t think is true).

The results for distinguishing invasive vs noninvasive melanomas is exciting, but the following statement is an over-reach and should be omitted: “Knowing the nature of the lesion 429 may impact the performance of lymphadenectomy or biopsy of sentinel lymph nodes dur- 430 ing primary surgery or the early use of additional treatments such as immunotherapy or 431 molecularly targeted therapy [54,55].“ as one would not expect that the well established histopathologic criteria to recommend sentinel node sampling, or staging imaging that might impact adjuvant therapy, would have a place in management algorithms.

Regarding limitation “disturbed by a large number of hairs in 448 the preparation or ulcers and other damage within the lesion” — please comment on the exclusion of lesions with hairs (since hairs cover much of the body) and ulcer (since ulceration is a common feature of aggressive melanomas).
Do you also expect the limitation of user training/expertise as noted for other noninvasive techniques?

Overall a very promising publication; would defer to statistics expert regarding power of the study given low numbers and the exclusion criteria.  Looking forward to bigger studies in the future!

Author Response

Dear Reviewer

Thank You for Your hard and valuable work, that let us to improve our manuscript. We added following corrections:

Thank you for sharing this analysis “Fractal Dimension Analysis of Melanocytic Nevi and Melanomas in Normal and Polarized Light – A Preliminary Report.“

For introduction it is appropriate to mention noninvasive diagnostics such as GEP, tape stripping which have promising data for adult pigmented lesions.

We added the paragraph as suggested.

It would be helpful to provide sample sensitivity/specificity for the approaches introduced, for comparison.

We added the data as suggested.

Please revise for clarity, am not sure what you’re saying: “Artificial intelligence and machine learning may allow in the future to reach specificity and sensitivity of the test not available for a clinical examination.”

We changed the sentence to be more clear.

The authors have acknowledged that they have already excluded a number of pigmented lesions that might be challenging to analyze - presumably this would include Spitz nevi and atypical Spitz tumors, nodular melanomas, etc.  There is often diagnostic discrepancy between severely dysplastic nevi and melanoma in situ — please provide a comparison of these two diagnoses specifically and whether they can be distinguished.  Also please provide overall information for melanomas (average, st dev for depth, mitoses, ulceration, etc.)

Jesteś w stanie to sprawdzić? Jakby wyszła różnica nieistotna statystycznie to mamy odpowiedź:

In our research group, we had too few Spitz nevi and atypical Spitz tumor cases to perform a relevant analysis. We excluded nodular melanomas due to their 3D structure, impossible to imaging in the contact photography mode performed with our dermoscopic lens. We also wanted to compare the lesions very similar to each other in the clinical examination, therefore we limited our analysis to the lesions described in the materials and methods. We performed analysis between dysplastic nevi and melanomas in situ, but it showed no statistical difference. However, we will be pleased to explore these differences in the future, perhaps with other image parameters as well. We have provided the requested information. Melanomas with ulceration was excluded from the study due to disturbance in lesion surface morphology, which could affect the results

In discussing novelty and exciting results, the paper is written in a way that suggests that most dermatologists are not using dermoscopy with polarized light (which I don’t think is true).

We changed the tone of our work as such an impression was not our intention. We are aware that this fact is commonly known by clinicians, but we believe that in the era of evidence based medicine it is useful to have quantifiable and measurable results proving the validity of this thesis, as a reference point for other scientific papers or textbooks for medical students

The results for distinguishing invasive vs noninvasive melanomas is exciting, but the following statement is an over-reach and should be omitted: “Knowing the nature of the lesion 429 may impact the performance of lymphadenectomy or biopsy of sentinel lymph nodes dur- 430 ing primary surgery or the early use of additional treatments such as immunotherapy or 431 molecularly targeted therapy [54,55].“ as one would not expect that the well established histopathologic criteria to recommend sentinel node sampling, or staging imaging that might impact adjuvant therapy, would have a place in management algorithms.

We changed this sentence, our intention was not to change the treatment algorithm, but if possible, faster implementation of individual steps algorithms. Please let us know if it is acceptable in this form, if the statement is still exaggerated, we will remove it from our work

Regarding limitation “disturbed by a large number of hairs in 448 the preparation or ulcers and other damage within the lesion” — please comment on the exclusion of lesions with hairs (since hairs cover much of the body) and ulcer (since ulceration is a common feature of aggressive melanomas).
Do you also expect the limitation of user training/expertise as noted for other noninvasive techniques?

The hair restriction applied to the area of ​​the hairy scalp due to the fact that the surface of the hair covers a significant part of the surface and circumference of the lesion, which may interfere with the analysis result and lead to the results being distorted. A small amount of hair covering other areas of the body, representing a low percentage of the preparation surface area, is acceptable. We have clarified the description in study limitations. The presence of ulcers and wounds on the surface of the lesion disturbs the structure of its surface, which is the basis of the analysis used, therefore lesions with damaged surface should not be the subject of a comparative analysis. Moreover, ulcerations on the surface of the lesion, as noted, are characteristic of malignant lesions of high aggressiveness, so they are usually a clear indication for the surgical removal of the lesion, not for its non-invasive diagnosis.
Since the subject of the analysis are photos with the use of a dermoscopic lens operating in the contact mode, which always ensures the same exposure parameters for the obtained photos, collecting images requires only minimal operator skills, including perpendicular setting of the lens and pressing the camera button. The implementation of the proposed analyzes in the form of CAD system could completely exclude the operator expertise resulting in the system completely independent of the operator.

Overall a very promising publication; would defer to statistics expert regarding power of the study given low numbers and the exclusion criteria.  Looking forward to bigger studies in the future!

Currently, our work is focused on the search for image parameters differentiating various types of pigmented lesions. Of course, our main goal will be to create a system to automatically diagnose these lesions, so we are going to fulfill this request.

Best regards,

Authors.

Round 2

Reviewer 1 Report

The authors corrected most of the comments and mistakes and also made explanations on some of the comments. Some of these comments were related to the results of statistical data analysis in software Statistics. Since many readers can use other statistical packages, it is better to describe data processing in more detail. Some important comments remained insufficiently clarified.

Comments:

1. Comment 3 (2.4.1. Preparing Images for Shape Analysis) was apparently written obscurely and the mistake still needs to be corrected.

In the article:

In line 220-221: “The grayscale image was set as a bottom layer, and the inverted image was  set as an upper  layer.”

In line 222: “One was added to the value of each pixel of the upper layer…”

Formula 1 and line 226-227: “…I – pixel value of the upper-layer image, M – pixel value of the lower-layer image.”

Figure 2: “Stack images into two layers " I" - lower layer, "M " - upper layer”.

Thus in different points: “I –upper-layer” and “I - lower layer”, “M - lower layer” and “M - upper layer”.

2. Comment 4. The authors have given a detailed explanation, but it is better to add a boundary criterion to the text: “to be selected as near to the lesion as possible, and it must not contain melanin spots”

3. Comment 5, 19. The authors have given very good explanations! However, this is not in the new version of the article, for example, in the discussion section. Perhaps the approach of the authors “to extract multiple ROIs from larger lesions” introduced distortions in the data sample and did not allow finding statistically significant differences in the studied groups (as in other works). Perhaps the sample was small, as the authors wrote in the discussion. It is useful to include the authors' explanation to comment 19 in the discussion section with a note of the limitations of the approach “to extract multiple ROIs from larger lesions”.

The authors are well done! You are doing a useful job!

Author Response

Dear Reviewer,

Thank You for valuable comments. We prepared following changes:

  1. Comment 3 (2.4.1. Preparing Images for Shape Analysis) was apparently written obscurely and the mistake still needs to be corrected.

In the article:

In line 220-221: “The grayscale image was set as a bottom layer, and the inverted image was  set as an upper  layer.”

In line 222: “One was added to the value of each pixel of the upper layer…”

One was added to the value of each pixel of the upper layer (to prevent dividing by zero). – it is correct.

Formula 1 and line 226-227: “…I – pixel value of the upper-layer image, M – pixel value of the lower-layer image.”

It was corrected: E – final pixel value, x, y – coordinates of each pixel of an image, I – pixel value of the lower layer image, M – pixel value of the upper layer image.  

Figure 2: “Stack images into two layers " I" - lower layer, "M " - upper layer”.

It is correct.

Thus in different points: “I –upper-layer” and “I - lower layer”, “M - lower layer” and “M - upper layer”.

All mismatched was corrected: I – lower layer, M – upper layer.

  1. Comment 4. The authors have given a detailed explanation, but it is better to add a boundary criterion to the text: “to be selected as near to the lesion as possible, and it must not contain melanin spots”

We added the sentence as suggested.

  1. Comment 5, 19. The authors have given very good explanations! However, this is not in the new version of the article, for example, in the discussion section. Perhaps the approach of the authors “to extract multiple ROIs from larger lesions” introduced distortions in the data sample and did not allow finding statistically significant differences in the studied groups (as in other works). Perhaps the sample was small, as the authors wrote in the discussion. It is useful to include the authors' explanation to comment 19 in the discussion section with a note of the limitations of the approach “to extract multiple ROIs from larger lesions”.

We added explanations in suggested sections. Perhaps multiple ROIs in fact didn't allow for statistical difference, but preventing bias of intentional choosing seems very important in this study, especially if this parameter would be used in automatic diagnostic systems, as inteded - the algorithm has to analyze the whole area of the lesion, and only one, randomly taken, small sample from larger lesion could lead to false diagnosis

Best regards,

Authors.

This manuscript is a resubmission of an earlier submission. The following is a list of the peer review reports and author responses from that submission.

Round 1

Reviewer 1 Report

Low evidence for clinical practice.

Reviewer 2 Report

My comments are attached in separate file.

Reviewer 3 Report

Dear authors, I read with interest and pleasure your paper entitled: "Fractal Dimension Analysis in Diagnosing Skin Pigmented Lesions in Normal and Polarized Light - A Preliminary Report ". In this paper, the authors aim to apply fractal analysis to pigmented lesions of the skin, divided respectively into benign nevi, dysplastic nevi and malignant melanoma. and the methodology is up to par with some good technical solutions. Here are my suggestions to improve your work.

Major Comment

Line 57-63: this paragraph is well written, but the authors wrote 7 lines (one paragraph!) without a single bibliographic citation on the histopathology and histopathological diagnosis of malignant melanoma.

I suggest to the authors to add some very recent papers that can be the background to the statements made by them regarding the histopathological diagnosis of melanoma. Here are the suggested works:

  1. Cazzato G, Arezzo F, Colagrande A, Cimmino A, Lettini T, Sablone S, Resta L, Ingravallo G. "Animal-Type Melanoma/Pigmented Epithelioid Melanocytoma": History and Features of a Controversial Entity. Dermatopathology (Basel). 2021 Jul 5;8(3):271-276. doi: 10.3390/dermatopathology8030033. PMID: 34287308; PMCID: PMC8293039.
  2. Bobos M. Histopathologic classification and prognostic factors of melanoma: a 2021 update. Ital J Dermatol Venerol. 2021 Jun;156(3):300-321. doi: 10.23736/S2784-8671.21.06958-3. Epub 2021 May 13. PMID: 33982546.
  3. Colagrande A, Cimmino A, Liguori G, Lettini T, Serio G, Ingravallo G, Marzullo A. Atypical Fibroxanthoma-Like Amelanotic Melanoma: A Diagnostic Challenge. Dermatopathology (Basel). 2021 Jan 12;8(1):25-28. doi: 10.3390/dermatopathology8010004. PMID: 33445655; PMCID: PMC7838949.
  4. Lospalluti L, Colagrande A, Cimmino A, Romita P, Foti C, Demarco A, Arezzo F, Loizzi V, Cormio G, Sablone S, Resta L, Rossi R, Ingravallo G. Dedifferentiated Melanoma: A Diagnostic Histological Pitfall-Review of the Literature with Case Presentation. Dermatopathology (Basel). 2021 Oct 15;8(4):494-501. doi: 10.3390/dermatopathology8040051. PMID: 34698090; PMCID: PMC8544555.
  5. Wilson ML. Histopathologic and Molecular Diagnosis of Melanoma. Clin Plast Surg. 2021 Oct;48(4):587-598. doi: 10.1016/j.cps.2021.05.003. Epub 2021 Jul 2. PMID: 34503719.
  6. Demarco A, Lospalluti L, Arezzo F, Resta L, Ingravallo G. The Great Mime: Three Cases of Melanoma with Carcinoid-Like and Paraganglioma-Like Pattern with Emphasis on Differential Diagnosis. Dermatopathology (Basel). 2021 May 13;8(2):130-134. doi: 10.3390/dermatopathology8020019. PMID: 34068376; PMCID: PMC8161759.

Line 152: the authors claimed :” significant hair presence”: what do they mean? Were pigmented skin lesions of the scalp excluded from this study? And if so, could this not become a bias? Please explain!

Line 152: “those located in an area other than the skin were 152 excluded from the study”: ??

Line 155: “The mean age of patients was 53, and the 155 median was 50, ranging from 14 to 90”: what? Years? Please, add them!

Methods and Results: OK!

Discussion: The Discussion section is well structured, with precise references to works already published. In my opinion the paper could become even more captivating and citable by obtaining information relating to the possible application of artificial intelligence algorithms in Dermatology and Dermatopathology, obtaining a paragraph that can allow readers to make a further "leap forward" in reading this work.

Minor Comments:

Reference n’9: please add the title of paper.

Reviewer 4 Report

The authors applied fractal dimension analysis to the examination of pigmented skin lesions. Although their idea is interesting, the presentation of the results is very poor. The quality of English writing is also unsatisfactory. Moreover, they do not mention to the discrimination ability between benign and malignant lesion, which would be the most interesting for the readers.

Major points

  1. Page 3 Line 131: Mention to the purpose of your study instead of listing null hypotheses.
  2. Page 6 Line 242: The result of Shapiro-Wilk test for each variable should be described in the Results section.
  3. Page 7 Line Table 1: Describe the subtypes of malignant melanoma: superficial spreading, nodular, acral lentiginous, or mucosal. In addition, the proportion of the subjects whether in situ or advanced should be reported.
  4. Page 7 Table 2: Description of the results are very redundant. Tables 2-4 can be integrated into a single table. Information for “Difference” and “SD Difference” is not necessary. In addition, I recommend the mean, the SD, and the results of Kruskal-Wallis test should be illustrated as bar graphs. By the way, isn’t it Mann-Whitney’s U test?
  5. Page 8 Table 4: Tables 4-5 can also be integrated. Adding bar graphs would be fine, too. Anyway, I am still not sure whether the authors conducting two-group comparison or multi-group comparison.
  6. Page 8 Table 6: Correlation between the variables should be demonstrated in scatter plots.
  7. Page 9 Table 7: Again, information included in the tables 7-8 should be displayed more concisely.
  8. Page 9 Table 9: Integrate the tables 9-10 into a single table.
  9. Discrimination ability of FD between MM and the others, or MM in situ and advanced MM should be examined more in detail (i.e. logistic analyses or ROC curve analyses.) This would be the most interesting for the readers.
  10. In the first paragraph of the Discussion section, please summarize your findings in the present study and its implication.

Minor points

  1. Page 4 Line 142: It would be informative if you illustrate the recruitment process in a flowchart.
  2. Page 4 Line 147: Explain “the Skin Cancer Unit protocol” in detail.
  3. Page 5 Line 195: Is this operation your original or common in the field of analyzing pigmented skin lesions?
  4. Displaying representative photos with their FD value would be more informative.